



# Microphysical Characteristics of Frozen Droplet Aggregates from Deep Convective Clouds

Junshik Um[1,2], Greg M. McFarquhar[2,3,4], Jeffrey L. Stith[4], Chang Hoon Jung[5], Seoung Soo Lee[6], Ji Yi
Lee[7], Younghwan Shin[8], Yun Gon Lee[9]

[1]Department of Atmospheric Sciences, Pusan National University, Busan, South Korea
[2]Cooperative Institute for Mesoscale Meteorological Studies, University of Oklahoma, Norman, Oklahoma, USA
[3]School of Meteorology, University of Oklahoma, Norman, Oklahoma, USA
[4]National Center for Atmospheric Research, Boulder, Colorado, USA
[5]Department of Health Management, Kyungin Women's University, Incheon, South Korea
[6]Earth System Science Interdisciplinary Center, College Park, Maryland, USA
[7]Department of Environmental Engineering, Chosun University, Gwangju, South Korea
[8]Department of Agricultural and Biological Engineering, University of Illinois, Urbana, Illinois, USA
[9]Department of Atmospheric Sciences, Chungnam National University, Daejeon, South Korea

*Correspondence to*: Junshik Um (scientistum@gmail.com)

**Abstract.** During the 2012 Deep Convective Clouds and Chemistry (DC3) experiment the National
Science Foundation/National Center for Atmospheric Research Gulfstream-V (G-V) aircraft sampled the
upper anvils of two storms that developed in eastern Colorado on 6 June 2012. A cloud particle imager
(CPI) mounted on the G-V aircraft recorded images of ice crystals at altitudes of 12.0 – 12.4 km and $T= -$
$61 - -55$ °C. A total of 22,393 CPI crystal images were analyzed, all with maximum dimension $D_{max} <$
433 μm with an average $D_{max}$ of 80.7±45.4 μm. The occurrence of well-defined pristine crystals (e.g.,
columns and plates) was less than 0.04% by number. Single frozen droplets and frozen droplet aggregates
(FDAs) were the dominant habits with fractions of 73.0% (by number) and 46.3% (by projected area),
respectively. The relative frequency of occurrence of single frozen droplets and FDAs depended on
temperature and position within the anvil cloud.

    A new algorithm that uses the circle Hough transform technique was developed to automatically
identify the number, size, and relative position of element frozen droplets within FDAs. Of the FDAs,
42.0% had two element frozen droplets and the frequency of occurrence decreasing as the number of
element frozen droplets increasing with the average of 4.70±5.0 element frozen droplets. Based on the
number, size, and relative position of the element frozen droplets within the FDAs, possible three-
dimensional (3-D) realizations of FDAs were generated and characterized by two different shape
parameters, the aggregation index (*AI*) and the fractal dimension ($D_f$), that describe 3-D shapes and link to
scattering properties with an assumption of spherical shape of element frozen droplets. The *AI* of FDAs is
seen to decrease with an increase in the number of element frozen droplets, with larger FDAs with more
element frozen droplets have more compact shapes. It is shown that $D_f$ of FDAs is about 1.20–1.43



smaller than that of BC aggregates (1.53–1.85) determined in previous studies. Such smaller $D_f$ of FDAs indicates that FDAs have more linear chain-like branched shapes than the compact shapes of BC aggregates. Determined morphological characteristics of FDAs along with the proposed reconstructed 3-D representations of FDAs in this study have important implication to improve the calculations of the

microphysical (e.g., fall velocity) and radiative (e.g., asymmetry parameter) properties of ice crystals in upper anvil clouds.

## 1 Introduction

Deep convective systems, such as thunderstorms and mesoscale convective systems (MCSs), play an important role in Earth's climate system, for example, by conveying ice crystals to the upper troposphere

and lower stratosphere, redistributing latent heat, controlling precipitation, and regulating the Earth's radiation budget (Jensen et al., 1996; Stephens 2005; de Reus et al., 2009; Frey et al., 2011; Feng et al., 2011, 2012; Gayet et al., 2012; Taylor et al., 2016). Clouds formed by deep convection show several distinct features. Vigorous turrets associated with deep convection generate intense precipitation that influences the hydrological cycle and large anvil shields that modulate radiation due to their extensive

spatial and temporal coverage (Feng et al., 2011; 2012; Wang et al., 2015). Overshooting tops associated with strong updrafts are responsible for stratosphere-troposphere exchange (Homeyer et al., 2014; Frey et al., 2015) and can be an indicator of a severity of thunderstorm (Proud, 2015).

Clouds formed by deep convection have three thermodynamic phases: liquid, mixed, and ice. The cloud particles also have different shapes, sizes, and concentrations that vary in the horizontal and vertical

causing horizontal and vertical variability in radiative properties. For example, precipitating cores of tropical convective clouds reveal a negative impact on radiation balance, whereas non-precipitating anvils have a positive impact (Hartmann and Berry, 2017). A numerical simulation (Fu et al., 1995) showed that the spatial extent of an anvil cloud is influenced by moisture advection from the convective turret, radiative effects, and small-scale convection occurring within the anvil (Lilly, 1988). The relationships

between the spatial and temporal coverage of convectively generated clouds and their radiative impact are still not well understood and affect the representation of cloud feedbacks in numerical models (Bony et al., 2015; 2016; Hartmann 2016; Hartmann and Berry 2017).

Despite the high height of the tropopause and the remote regions where some of these cloud systems occur, there have been in-situ measurements of the microphysical and scattering properties of ice

crystals in anvil tops (e.g., Heymsfield 1986; McFarquhar and Heymsfield, 1996; Stith et al., 2002; 2004; 2014; 2016; Connolly et al., 2005; Gallagher et al., 2005; Heymsfield et al., 2005; May et al., 2008; Jensen et al., 2009; Lawson et al., 2010; Frey et al 2011; Gayet et al., 2012; Barth et al., 2015; Jensen et





al., 2016). Although in-situ aircraft measurements have some limitations since crystals are not observed where they form (Um et al., 2015), they along with laboratory experiments (e.g., Bailey and Hallett, 2004; 2009) provide information on how crystal habit varies with temperature and humidity.

One distinct characteristic of anvil clouds is the frequent occurrence of plate type crystals and their aggregates, which is different from ice crystals found in non-convective cirrus, where bullet rosettes and their aggregates are most common (McFarquhar and Heymsfield et al, 1996; Stith et al., 2002; Lawson et al., 2003; Connolly et al., 2005; Um and McFarquhar, 2009; Järvinen et al., 2016). Since plate type crystals form at warmer temperatures (Bailey and Hallett, 2004; 2009) than typical temperatures at anvil tops, these plate crystals must form at lower altitudes and be transported to upper altitudes by

convection. It has been hypothesized that the "chain–like" shaped aggregates frequently observed in convective clouds may be produced by high electric fields within clouds (Saunders and Wahab, 1975; Stith et al., 2002; 2004; Lawson et al., 2003; Connolly et al., 2005; Um and McFarquhar, 2009). These shapes differ from aggregates observed in non-convective cirrus where aggregates of bullet rosettes are more common (Um and McFarquhar, 2007) and chain-like structures are not commonly seen.

Another unique feature of ice crystals in deep convective clouds is the high concentration of small frozen droplets (Gayet et al., 2012; Baran et al., 2012; Stith et al., 2014). These are no doubt generated from the freezing of supercooled droplets that has been observed at temperatures as low as −37.5 °C in deep continental convective clouds (Rosenfeld and Woodely, 2000). Below this temperature homogeneous freezing occurs in the strong updrafts producing high concentrations of quasi-spherical

frozen droplets with maximum dimensions $D_{max} < {\sim}50$ μm (Heymsfield and Sabin, 1989; Phillips et al., 2007; Heymsfield et al., 2009). However, instrumental uncertainties associated with the generation of artificially high concentrations of small ice crystals from shattering of large ice crystals on the tips of in-situ probes (Field et al., 2003; 2006; McFarquhar et al., 2007; Korolev et al., 2011; Lawson, 2011; Jackson and McFarquhar, 2014; Jackson et al., 2014; Korolev and Field, 2015) cast some doubt on the

exact concentration of these small crystals. Knowledge about the concentrations, shapes, and scattering properties of these small crystals at cloud top is crucial for satellite retrievals that rely on visible and near-infrared wavelengths because of their strong influence on cloud reflectance (e.g., Stephens et al., 1990; McFarquhar et al., 1999; Yang et al., 2001).

        Although not plentiful, there are some observations of the shapes of these small ice crystals at the

30 tops of anvils and convective towers. Gayet et al. (2012) reported up to 70 cm$^{-3}$ of frozen droplets and their aggregates with chain–like shapes in the overshooting tops of a continental deep convective cloud at $T{\sim}-58$ °C. A mean effective diameter of 43 μm, maximum particle size of ${\sim}300$ μm, and an asymmetry parameter of ${\sim}0.776$ were observed in the very dense cloud tops. Single frozen droplets and frozen droplet aggregates (FDAs) were also observed in mid-latitude continental convective clouds during the 2012



Deep Convective Clouds and Chemistry (DC3) experiment (Barth et al., 2012; Stith et al., 2014). Based on these observations, the occurrence of single frozen droplet and FDAs as a function of position within the anvil cloud and of updraft velocity was determined (Stith et al., 2014). A positive correlation between the frequency of occurrence of FDAs and the level of nitrogen oxide ($NO_x$) produced by lightning

suggested that a strong electric force in the updrafts was playing a role in the formation of the FDAs (Stith et al., 2016). On the other hand, these FDAs have not been observed in tropical or maritime convective clouds where updrafts are not as strong. Aggregates of faceted crystals (e.g., plate) are more common in these systems (Stith et al., 2002; 2004; Lawson et al., 2003; Connolly et al., 2005; Um and McFarquhar, 2009; Gallagher et al., 2012) because droplets originating at cloud base are less likely to

reach the homogeneous freezing level of –38 °C (Heymsfield et al., 2009).

     The radiative properties (e.g., albedo) of convective cloud systems depend strongly on both the concentrations and shapes of crystals in the anvil-cloud layer. In order to better understand the role of continental convective clouds in Earth's radiation budget, the fractional contributions of different habits must be quantified, and the scattering properties of the habits determined. This is complicated by a couple

of issues. First, several idealized crystal models representing shapes of small crystals have been proposed (McFarquhar et al., 2002; Yang et al., 2002; Nousiainen and McFarquhar, 2004; Nousiainen et al., 2011; Um and McFarquhar, 2011; 2013; Järvinen et al., 2016), but it is not known which best characterizes the shapes. Second, few in-situ aircraft observations of continental convective clouds have been made due to their high altitudes and the difficulty of flying through or near strong updrafts. In this study, 22,393

crystals imaged by a Cloud Particle Imager (CPI) on 6 June 2012 in anvil clouds over eastern Colorado during DC3 are analyzed to determine the morphological properties of single frozen droplets and FDAs (e.g., size and number of element) and their radiative impacts. Although previous studies (Gayet et al., 2012; Baran et al., 2012; Stith et al., 2014; 2016) analyzed FDAs observed in continental deep convective clouds, the dimensions and three-dimensional (3-D) shapes of FDAs that are important for radiative

implication were not determined as is done in this study.

     The remainder of this paper is organized as follows. Section 2 summarizes the in-situ aircraft measurements made during DC3. In Section 3, a habit classification scheme that distinguishes FDAs from other crystals is introduced along with the methodology used to identify the number and size of element frozen droplets within FDAs. The morphology of FDAs and their reconstructed 3-D shapes are shown in

Section 4. Two different parameters that describe the 3-D shapes of aggregate particles, fractal dimension and aggregation index, are also introduced. Further, the characteristics of the shapes of FDAs are compared against those of black carbon aggregates in Section 4. The significance of this study and concluding remarks are made in Section 5.



## 2 Cloud probes and in-situ measurements

### 2.1 Cloud probes

The 2012 DC3 experiment investigated the impacts of deep mid-latitude continental convective clouds on upper tropospheric chemistry and composition in the Midwest of the United States (Barth et al., 2015).

The National Science Foundation (NSF)/National Center for Atmospheric Research (NCAR) Gulfstream-V (G-V), the National Aeronautics and Space Administration (NASA) DC-8, and the Deutsches Zentrum für Luft- und Raumfahrt (DLR) Falcon aircraft were deployed during DC3.

In this study, in-situ measurements were acquired from the G-V equipped with a Stratton Park Engineering Company Inc. (SPEC) 3V-CPI, a Cloud Droplet Probe (CDP, manufactured by Droplet

Measurement Technologies, DMT), and a specially modified Particle Measuring Systems (PMS) optical array probe (2DC), which uses high-speed electronics and a 64-element 25 µm-resolution diode array in order to shadow particles at the sampling speeds of the G-V. The 3V-CPI provided high-resolution (i.e., 2.3 µm) images of ice crystals with sizes up to ~2300 µm at 400 frames per second. The CDP determined the number distribution function $N(D_{max})$ and total concentration ($N_{CDP}$) of ice crystals with $D_{max}$ between

2 and 50 µm from the amount of light forward scattered. The 2DC optical array probe with 64 elements and 25 µm resolution measured $N(D_{max})$ for $D_{max}$ < 1550 µm. Other details on the G-V instrumentation are provided in Stith et al., (2014).

The shattering of large cloud particles on the shrouds, tips, or inlets of cloud probes can cause artificial increases in in-situ measured concentrations of small particles. The impacts of shattering must

thus be prevented or removed (Field et al., 2003; 2006; McFarquhar et al., 2007; Korolev et al., 2011; Lawson 2011; Jackson and McFarquhar, 2014; Jackson et al., 2014; Korolev and Field, 2015). The CDP used during DC3 did not have a shroud, and thus shattering is not expected to be substantial (Stith et al., 2014). Anti-shattering tips (Korolev et al., 2011) were installed on the 2DC, and post-processing methods of removing particles with small interarrival times (Field et al. 2003; 2006) were applied. 2DC

measurements of only particles with $D_{max}$ > 100 µm were used in this study due to a poorly defined depth of field for smaller particles (e.g., McFarquhar et al., 2017). Thus, the CDP ($N_{CDP}$) and 2DC ($N_{2DC>100}$) concentrations are used to identify the presence of small ($D_{max}$ < 50 µm) and large ($D_{max}$ > 100 µm) particles, respectively. Information about particle shape was obtained from the CPI component of the 3V-CPI. But, only CPI images with focus > 20 were used (McFarquhar et al., 2013), and analysis of multiple

particles on the same frame was not performed since Um and McFarquhar (2011) suggested they might be shattered artifacts.



## 2.2 Anvil observations

During the 6 June 2012 flight, the G-V sampled the upper anvils of two storms that developed in eastern Colorado between 221000 and 223000 UTC near the CHILL radar (40.45 °N and 104.64 °W). Two storms were aligned north-south, separated by ~70 km, and had similar size and intensity based on the

next generation weather radar (NEXRAD) images (see Figs. 1-3 of Stith et al. (2014)). Examples of CPI crystal images sampled during this flight are shown in Fig. 1. They were mainly (> 93% by number) single frozen droplets with quasi-circular shapes and their aggregates (i.e., FDAs). This is consistent with the analysis of Stith et al. (2014), who showed that these upper anvil regions were primarily composed of frozen droplets with differing degrees of aggregation, with FDAs being most frequent in the center and

lower regions of the upper anvil. More details about these two storms are discussed in Stith et al. (2014; 2016). The G-V flew two constant altitude runs during this period, at altitudes and temperatures of 12.0 – 12.4 km and –61 – –55 °C, respectively (see Fig. 2). This period is further segregated into three periods: 1: 221130–221455 UTC, 2: 221900–222340 UTC, and 3: 222355–222750 UTC (Table 1 and Fig. 2). These periods are selected in order to separate measurements in the north and south anvils and at different

temperatures. During time periods 1 and 2, the same anvil of the south storm was sampled with ~4 minute interval between the samples at two different temperatures ($T$~ –68.4 and –57.5°C, respectively), whereas period 3 sampled the north anvil at the higher temperature of –56.6 °C. In the next section, the frequency at which single frozen droplets and FDAs were observed is determined using a technique derived to identify the frozen droplet elements within FDAs.

## 3 Methodology

### 3.1 Ice crystal habit classification

Based on CPI crystal images obtained in tropical ice clouds, Um and McFarquhar (2009) developed a classification scheme to sort crystals into eleven habits: small, medium, and large quasi-spheres, columns, plates, bullet rosettes, aggregates of columns, aggregates of plates, aggregates of bullet rosettes, capped

columns, and unclassified. To represent other crystal habits commonly found in mid-latitude and Arctic clouds, the capability of sorting into more habits (i.e., dendrite, needle, aggregates of needles, and FDAs) has been added to the scheme (McFarquhar et al., 2017). Thus, this habit classification scheme now sorts crystals into 15 different categories in a quasi-automatic manner that requires some manual intervention.

       In this study, ice crystals classified as small (SQS), medium (MQS), and large quasi-spheres

(LQS) are regarded as single frozen droplets. The FDAs that occur near anvil tops are often classified as bullet rosettes, aggregates of bullet rosettes, or unclassified from the automated part of the algorithm, and



thus an additional manual check was necessary to confirm whether or not these crystals were FDAs. To be classified as FDAs, there must be at least two quasi-circular frozen droplets as elements. Habits that frequently occurred during the 6 June 2012 flight were single frozen droplets (i.e., SQS, MSQ, and LQS) and their aggregates (i.e., FDAs), whereas very few pristine shape crystals, such as plates, columns, and

bullet rosettes, were observed (see Table 1 and Fig. 3).

### 3.2 Technique to identify element frozen droplets

The circle Hough transform (CHT, Duda and Hart, 1972) detects circular objects in digital images and is one of many feature-extracting techniques that use the Hough transform (Hough 1962). Several variants of the Hough transform have been developed, such as, the fast Hough transform (Li et al., 1986), two-

stage CHT (Yuen et al., 1990), space saving approach CHT (Albanesi and Ferretti, 1990), and the phase-coding method (Atherton and Kerbyson, 1999). These techniques have been used to detect natural particles with circular shapes in digital images, such as, circular nanoparticles in transmission electron microscopy (TEM) images (Bescond et al., 2014; Mirzaei and Rafsanjani, 2017).

    Prior studies have used such techniques to identify the elemental or primary particles within black

carbon aggregates (e.g., Bescond et al., 2014; China et al., 2013), most of which are circular. Similar techniques can be applied to the FDAs observed near the tops of anvil clouds assuming the element frozen droplets have spherical shapes. The biggest difference between TEM and CPI images is that the quality of TEM images is, in general, better than that of the CPI images. A CPI image has an inhomogeneous background and debris or noise, such as impulse noise (i.e., salt-and-pepper noise), which

causes lower quality images. Thus, additional image-quality control was required before applying the CHT technique to the images. This was accomplished in a number of steps. First, a median filter that is a nonlinear digital filtering technique to remove noise is applied to the CPI images classified as FDAs. The 256-level gray-scale CPI images are then converted to binary images based on the average intensity of pixels to further remove background noise and debris. Figure 4 shows example images of CPI FDAs.

Two different CHT techniques, the two-stage CHT and phase-coding method, are then applied to the images to detect element circles (i.e., frozen droplets) as shown by the red circles in Figs. 4 and 5. Two different techniques are used because the performance of each technique varies depending on the CPI image being classified. The technique used for the subsequent analysis is chosen as that for which the projected area of the FDAs determined for the element frozen droplets identified by CHT technique (i.e.,

area determined by red lines in Fig. 5) best matches that for the original CPI image (i.e., area enclosed by green line in Fig. 5). For example, the phase-coding technique shows closer agreement to the imaged area for the FDAs shown in the top row of Fig. 5, while the two-stage CHT does for the FDAs shown in the bottom row. The number and size of the element frozen droplets within the FDAs were then determined



automatically. The relative positions (i.e., $x$ and $y$ coordinates) of the element frozen droplets were also identified.

## 4 Results

### 4.1 Frequency of occurrence of ice crystal habits

Figure 3 shows the normalized contribution of each habit to the total number (red) and to the total projected area (blue) of measured ice crystals during the three different time periods and integrated over the entire time period. For all time periods, single frozen droplets represented the dominant habit by number, whereas FDAs were dominant by projected area (see also Table 1). The fraction (by number) of single frozen droplets was 73.0% (84.1; 70.6; 71.2%) for all periods (period 1; period 2; period 3),

whereas the area fraction of FDAs was 46.3% (27.8; 49.6; 47.5%). The fraction of well-defined pristine ice crystals, such as plates and columns, was less than 0.04% by number and 0.12% by area for all time periods, whereas unclassified crystals represented 6.1% (3.9; 5.3; 7.5%) by number for all periods (period 1; period 2; period 3) and 13.5% (6.5; 10.9; 16.9%) by area. These fractions of unclassified crystals were lower than those obtained from anvil cloud in the Tropics (Um and McFarquhar, 2009) that showed more

than 22% and 37% contributions by number and area, respectively. The presence of small crystals with relatively simple habit distributions shown in this study indicates that the anvil clouds were sampled in early developing stage, which was verified by using radar observations (Stith et al., 2014; 2016).

       The average $D_{max}$ for all crystal habits was 80.7±45.4 µm for all periods, with the average $D_{max}$ of 68.4±37.1 µm during period 1 at $T = -60.0±1.2$ °C smaller than those of 72.4±42.9 µm and 84.4±48.8 µm

during periods 2 and 3 at $T = -57.5±0.3$ °C and $T = -56.6±0.5$ °C, respectively (Table 1). Table 1 also shows that the contributions of single frozen droplets and FDAs sampled in the south and north anvil in periods 2 and 3 are similar, whereas a much higher fraction of small frozen droplets was revealed in the south anvil during period 1 at a slightly lower temperature. For example, the fraction (by number) of single frozen droplets was 84.1% in period 1, and 70.6% and 71.2% in periods 2 and 3, respectively. The

fraction of FDAs in period 1 was 12.1% and 27.8% by number and projected area, respectively, substantially lower than those sampled in periods 2 and 3 (Table 1). Figure 2 shows the fraction of single frozen droplet, in general, decreased as the G-V penetrated into the center of anvil cloud (i.e., center of each gray shaded area shown on panels in Fig. 2), and then increased as it approached to the cloud edge (i.e., both sides of each gray shaded area shown on panels in Fig. 2) for all periods. This variation in

relative occurrence of small and large crystals is more distinctly seen by comparing the $N_{CDP}$ and $N_{2CD>100}$ shown in Fig. 2.





In summary, single frozen droplets and their aggregates dominated the upper anvil clouds sampled in-situ, with the relative frequency of occurrence of single frozen droplets and FDAs dependent on temperature and position within the anvil, consistent with the conceptual model proposed by Stith et al. (2014, Fig. 12) and further detailed in Stith et al. (2016, Fig. 9).

## 4.2 Morphology of single frozen droplets and FDAs

Among the 4,667 CPI images of FDAs, the CHT technique succeeded in identifying element frozen droplets for 4,356 FDAs, whereas it failed for 311 FDAs (6.66%). The number, size, and 2-D position of the element frozen droplets within the FDAs were thus determined automatically. Figure 6 (top panel) shows the frequency distribution of the number of element frozen droplets within FDAs. The average number of frozen droplets within FDAs is 4.70±5.0 and the FDAs with two element frozen droplets are dominant with a frequency of occurrence of 42.0%. This occurrence frequency gradually decreases with the number of element frozen droplets. The average and standard deviation of diameter of the determined element frozen droplets (blue) are shown as a function of the number of element frozen droplets (bottom panel of Fig. 6). The average and standard deviation of $D_{max}$ of the single frozen droplets (red) are also shown for comparison. Considering plausible errors (~±4.6 μm) in the identifying algorithm and the 2.3 μm CPI resolution, the average and standard deviation of diameter of the element frozen droplets (31.79±7.12 μm) are comparable with those of the $D_{max}$ of single frozen droplets (34.03±6.22 μm). The quasi-spherical shapes, non-pristine shapes, and similarity of single and element droplet sizes indicate that diffusional growth was likely not effective for the anvils sampled and the large ice crystals (i.e., FDAs) grew mainly through aggregation. They also indicate that the sampled anvil clouds are in early developing stage as verified by radar observations. The more complex structure of FDAs with an increase of number of elements may cause errors in the estimated size of the element frozen droplets. For example, the increase in the determined diameter of element frozen droplets as the number of element frozen droplets increases from 2 to 5 in Fig. 6 (bottom panel) may not be a physical effect, but rather caused by uncertainty in the methodology. The sizes of the element frozen droplets here (31.79±7.12 μm) are larger than those (15-20 μm) noted by Gayet et al. (2012). It is hard to determine the extent to which differences in methodology as opposed to physical differences in droplet sizes caused these differences because Gayet et al. (2012) did not specify how they determined frozen droplet size. But, despite the unavoidable errors in identifying the element frozen droplets due to the quality of the CPI images, the differences are large enough to suggest physical differences.



### 4.3 Three-dimensional representations of FDAs and comparison with black carbon aggregates

To determine microphysical (e.g., fall velocity) and scattering (e.g., asymmetry parameter) properties of cloud particles required for models, idealized 3-D models of the crystals are needed. But, cloud particle images recorded by cloud probes are silhouettes (i.e., 2-D images) of 3-D cloud particles (Nousiainen and

McFarquhar, 2004). Retrieving the 3-D shapes of cloud particles based on the recorded silhouettes is difficult, especially for non-spherical ice crystals that have non-pristine shapes. It is easier to reconstruct 3-D shapes of well-defined pristine crystals, such as columns and plates. For example, an iterative approach to retrieve the 3-D shapes of bullet rosette crystals was developed (Um and McFarquhar, 2007). Assuming that the element crystals all had the same shape (e.g., plates), the 3-D shapes of more complex

crystal aggregates (e.g., aggregates of plates) have also been reconstructed from crystal silhouettes (Um and McFarquhar, 2009).

FDAs consist of at least two element frozen droplets whose shapes are assumed to be spheres even though the elements are in fact quasi-spherical, meaning they have some departures from a spherical shape. The number, size, and 2-D position of the element frozen droplets within the FDAs were

determined from the CPI images as explained in Section 3.2. Using this information, the 3-D shapes of FDAs are reconstructed for given the 2-D silhouette (i.e., CPI image) with the following assumptions:

- Element frozen droplets of FDAs are spheres.
- There is no overlap between the elements of the frozen droplets.
- The maximum number of contacting points of an element frozen droplet with other frozen

droplets is 2.

Since the relative positions (i.e., $x$ and $y$ coordinates) and sizes of the element frozen droplets are known, the reconstruction problem becomes one of stacking spheres with varying combinations of a vertical ($z$) coordinate. The above assumptions reduce the number of possible 3-D realizations of FDAs for a given 2-D projection so that a maximum $2^{n_p-2}$ 3-D realizations are possible, where $n_p$ is the number of element

frozen droplets. For example, for FDAs with five element frozen droplets (i.e., $n_p$=5), a maximum of eight different 3-D realizations are possible, while a maximum of 256 3-D realizations is possible for FDAs with 10 elements. Figure 7 shows six different example 3-D realizations of the FDA shown in the top-left panel of Fig. 4. Since this FDA has 12 element frozen droplets, theoretically a total of 1,024 3-D realizations are possible. However, such FDAs with large number of element frozen droplets usually have

much less 3-D realizations because of the above-mentioned assumptions, in particular due to the no overlap assumption.

As the number of element frozen droplets increases, the number of possible 3-D realizations also increases. For FDAs with 20 element frozen droplets, a maximum of 262,144 different 3-D realizations is



possible. Considering all 262,144 3-D realizations of FDAs is impractical for calculations of single-scattering properties. Thus, parameters that characterize the 3-D shapes of particles and link the 3-D shapes to scattering properties are required. Um and McFarquhar (2009) used several parameters, such as the aggregation index ($AI$), area ratio, and normalized projected area, to characterize the 3-D shape and to

link to the scattering properties of aggregates of plate crystals. The motivation for the use of the $AI$ is that the asymmetry parameter ($g$) of aggregates of plates was previously seen to increase with $AI$ (Um and McFarquhar 2009). In this study, a similar approach is adapted and the $AI$ is defined as

$$AI = \frac{\sum_{i=1}^{n_p} \sum_{j=1}^{n_p} D_{ij}}{Max\left(\sum_{i=1}^{n_p} \sum_{j=1}^{n_p} D_{ij}\right)}, \qquad (1)$$

where $D_{ij}$ is the distance between the center of frozen droplet $i$ and that of frozen droplet $j$ and $n_p \geq 3$.

Physically, $AI$ represents the ratio of the sum of the distances between the centers of all frozen droplets compared to that when they all lie on a straight line. Thus, when all frozen droplets line on a straight line, $AI$=1, whereas smaller $AI$ implies a more compact shape. The $AI$ is calculated for every 3-D realization. Thus, there are 262,144 $AI$s for FDAs with 20 element frozen droplets. Since the 3-D shape complexity of a particle (i.e., $AI$) and its $g$ was previously shown to have a positive relationship for aggregates of plate

(Um and McFarquhar 2009), the maximum, minimum, and average $AI$ are calculated for a given FDA. The calculated maximum, minimum, and average $AI$ of FDAs as a function of the number of element frozen droplets (for $n_p>2$) are shown in Fig. 8. The $AI$ of FDAs decreases with $n_p$, which indicates that larger FDAs with more element frozen droplets have more compact shapes.

A fractal dimension ($D_f$) has been widely used to describe the 3-D shape or compactness of black

carbon (soot) aggregates (e.g., Bescond et al., 2014; China et al. 2013) and can be represented using the following scaling law:

$$n_p = k_f \left(\frac{R_g}{r_p}\right)^{D_f}, \qquad (2)$$

where $R_g$, $r_p$, and $k_f$ is the radius of gyration, average radius of elements frozen droplets, and fractal prefactor (or structural coefficient) (Mandelbrot 1982), respectively. The radius of gyration is represented

as

$$R_g^2 = \frac{\sum_i^{n_p} x_i^2 m_i}{\sum_i^{n_p} m_i}, \qquad (3)$$

where $m_i$ is the mass of $i^{th}$ element and $x_i$ is the distance between element $i$ and the center of mass of the FDA. The radius of gyration is an overall cluster radius. The chain-like shapes of FDAs observed near the tops of anvil clouds (e.g., Fig. 1) show similarity with those of black carbon (BC) aggregates (see Figs. 6–

8 in Lewis et al. (2009)). Thus, identifying fractal dimensions of FDAs and comparing them against those of BC aggregates is of great interest to calculate the scattering properties as a function of shape. In this



study, possible 3-D realizations of FDAs are represented using both $AI$ and $D_f$ with subsequent comparison to the shape of BC aggregates.

To provide an overview of the shapes of the FDAs, the $D_f$, $k_f$, $R_g$, and $r_p$ are calculated for three 3-D realizations of a FDA that represent the maximum, minimum, and average $AI$. For each 3-D realization

$R_g/r_p$ is plotted against $n_p$ in Fig. 9. Then, $D_f$ and $k_f$ are determined by fitting to Eq. 2. For a given $D_f$, $k_f$ represents the degree of compactness, with a smaller $k_f$ indicating less packing density (Lewis et al., 2009). The FDAs with the maximum, minimum, and average $R_g/r_p$ have $D_f$ ($k_f$) of 1.2083 (2.0998), 1.4329 (2.3864), and 1.4124 (2.0412), respectively. Smaller $D_f$ indicates a more linear shape with 1.0 indicating a perfectly linear shape and the compactness increases with $D_f$. Thus, FDAs with the maximum $R_g/r_p$ have

more linear (chain-like) shapes, while FDAs with the minimum $R_g/r_p$ have more compact shapes.

The $D_f$=1.8 and $k_f$=1.35 for BC aggregates determined in previous studies (Meakin, 1983; Kolb et al., 1983; Sorensen and Robert, 1997; Lattuada et al., 2003; Pierce et al., 2006; Heinson et al., 2012; Heinson and Chakrabarty, 2016) are shown in the purple dashed line in Fig. 9 for comparison. The $D_f$=1.85 and $k_f$=1.46 determined for ambient BC aggregates (black dashed line) and $D_f$=1.53 and $k_f$=2.40

for denuded BC aggregates (yellow dashed line) sampled from the Las Conchas fire (New Mexico, 2011) (China et al., 2013) are also shown in Fig. 9. The calculated fractal dimensions (1.20-1.43) of FDAs are smaller than those of BC aggregates (1.53-1.85), which indicates that FDAs have more linear branched shapes compared to the compact shapes of BC aggregates. Previous studies showed that $D_f$ of fresh BC aggregates is between 1.6 and 1.8 (e.g., Chakrabarty et al., 2006; Köylü et al., 1995; Sorensen, 2001;

Pierce et al., 2006; Heinson et al., 2012) and becomes larger than 1.8 with age of BC aggregates (Lewis et al., 2009). It was also shown that the scattering properties of BC aggregates depended heavily on their $D_f$ (e.g., Sorensen, 2001; Liu et al., 2008). Thus, it is required to characterize the 3-D shapes of FDAs for the accurate calculations of radiative properties.

There is a fundamental difference in the nature of the variables $AI$ and $D_f$ that describe the 3D

shapes of aggregates. The former is a 3-D shape indicator of individual aggregates, whereas the latter is an indicator for a group of aggregates. Each parameter has advantages and disadvantages. For example, $AI$ is useful to describe the 3-D shape of individual aggregates and their scattering properties. Since the $D_f$ is intended to describe a group of aggregates, a statistically significant number of samples is required to determine meaningful values and they should not be used to describe individual aggregates. Though $AI$

and $D_f$ cannot be compared, a comparison between $AI$ and $R_g/r_p$ is possible. Comparisons between $AI$ and $R_g/r_p$ of FDAs with 10 (left) and 20 (right) element frozen droplets are shown in Fig. 10. The CPI images of FDAs are embedded in each panel. For the FDA with 10 elements a total 127 3-D realizations are possible, whereas a total 65,535 3-D realizations are possible for the FDA with 20 elements shown in Fig. 10. The best fits illustrated in Fig. 10 show that $AI$ and $R_g/r_p$ of FDAs are positively correlated with high



correlation coefficients. For all FDAs with $n_p > 2$, an average $r$ of 0.942±0.094 is revealed. It indicates that the $AI$ and $R_g/r_p$ of aggregates are highly correlated and both parameters can be used to describe individual and/or a group of aggregates for the calculations of microphysical and radiative properties.

## 5 Summary and conclusions

During the 2012 Deep Convective Clouds and Chemistry (DC3) experiment the National Science Foundation/National Center for Atmospheric Research Gulfstream-V (G-V) aircraft sampled the upper anvils of two storms that developed in eastern Colorado on 6 June 2012. A cloud particle imager (CPI) mounted on the G-V aircraft recorded images of ice crystals at altitudes of 12.0 – 12.4 km and $T$= –61 to –55 °C. The G-V flew two constant altitude runs during this period that were segregated into three

periods according to which anvil and the temperature level sampled. A total of 22,393 CPI crystal images were analyzed, all with maximum dimension $D_{max}$ < 433 μm with an average $D_{max}$ of 80.7±45.4 μm. Dominant crystal habits observed during the 6 June 2012 flight were single frozen droplets and frozen droplet aggregates (FDAs, see Fig. 1). A new algorithm that uses the circle Hough transform technique was developed to automatically identify the number, size, and relative position of element frozen droplets

within FDAs and was applied to 4,667 FDAs. Using this information, the 3-D shapes of FDAs were reconstructed for given 2-D silhouettes (i.e., CPI images) and two different parameters describing the 3-D shapes of aggregate particles, the aggregation index ($AI$) and fractal dimension ($D_f$), were determined. The characteristics of the shapes of FDAs were compared against those of black carbon (BC) aggregates.  The anvil cloud selected for this study was an early-stage anvil associated with a strong continental storm and

appeared to provide conditions most favorable for the formations of frozen drops and FDAs, as other ice particle types were mostly absent in the location selected for study. Other anvils from DC3 exhibited FDA's as common ice particle types, but to a lesser extent than the cloud regions sampled here (Stith et al., 2014).

      The most important findings from this study are summarized as follows:

1. For all time periods, single frozen droplets represented the dominant habit by number, whereas FDAs were dominant by projected area. The fraction (by number) of single frozen droplets was 73.0% (84.1; 70.6; 71.2%) for all time periods (period 1; period 2; period 3), whereas the area fraction of FDAs was 46.3% (27.8; 49.6; 47.5%).

2. The fraction of well-defined pristine ice crystals (i.e., plates and columns) was less than 0.04% by number and 0.12% by area for all time periods, whereas unclassified crystals represented 6.1% (3.9;





5.3; 7.5%) by number for all periods (period 1; period 2; period 3) and 13.5% (6.5; 10.9; 16.9%) by area.

3. The high concentrations of small crystals (i.e., single frozen droplets) with relatively simple habit distributions shown in this study indicates that the anvil clouds were sampled in early developing stage as also verified by using radar data.

4. The relative frequency of occurrence of single frozen droplets and FDAs was dependent on temperature and position within the anvil, consistent with the conceptual model proposed by Stith et al. (2014; 2016). The fraction of single frozen droplets, in general, decreased as the G-V penetrated into the center of anvil cloud, and then increased as it approached to the cloud edge for all three periods.

5. The average number of element frozen droplets within FDAs is 4.70±5.0. The FDAs with two elements were dominant with a frequency of occurrence of 42.0%. This occurrence frequency gradually decreased with the number of element frozen droplets.

6. The average diameter of the element frozen droplets (31.79±7.12 μm) was comparable with that of single frozen droplets (34.03±6.22 μm). The quasi-spherical shapes, non-pristine shapes, and similarity of single and element droplet sizes indicate that diffusional growth was likely not effective and the large ice crystals (i.e., FDAs) grew mainly through aggregation.

7. The $AI$ of FDAs decreases with an increase in the number of element frozen droplets, which indicates that larger FDAs with more element frozen droplets have more compact shapes.

8. The calculated fractal dimensions of FDAs (1.20-1.43) in this study are smaller than those of BC aggregates (1.53-1.85), which indicates that FDAs have more linear branched shapes compared against the compact shapes of BC aggregates.

9. A strong positive relationship ($r$=0.942±0.094) between $AI$ and the ratio of radius of gyration $R_g$ to the average radius of element frozen droplets $r_p$) of FDAs is shown. Both parameters can be used to describe 3D shapes of aggregates and to link the scattering properties, especially $AI$ and $D_f$ for an individual and a group of aggregates, respectively.



The results of this study have important implication to improve the calculations of the microphysical (e.g., fall velocity) and radiative (e.g., asymmetry parameter) properties of ice crystals in upper anvil clouds, especially continental convective clouds. A subsequent study will calculate the single-scattering properties and fall velocities of FDAs using the morphological features and models of FDAs proposed here, which will have high impacts on clouds formed over U.S great plain and east Andes where strongest convection and electric field exist.

## Acknowledgements

This work was supported with funding from the National Science Foundation under Grant AGS 12-13311 and from the Advanced Study Program (ASP) at the National Center for Atmospheric Research. Part of this work was completed while GM was on sabbatical at NCAR. This research was supported by the National Strategic Project – Fine particle of the National Research Foundation of Korea (NRF) funded by the Ministry of Science and ICT (MSIT), the Ministry of Environment (ME), and the Ministry of Health and Welfare (MOHW) (NRF–2017M3D8A1092022). We would like to acknowledge operational, technical and scientific support provided by NCAR's Earth Observing Laboratory, sponsored by the National Science Foundation. CPI Imagery is available from (UCAR/NCAR, 2013) and low-rate G-V data are available from (UCAR/NCAR, 2103).

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




Table 1. Segregated time periods of 6 June flight and contributions (%) of crystal habit to the total number (total projected area) of ice crystals for given time period. Average and standard deviation of temperature ($T$), altitude, and maximum dimension ($D_{max}$) of ice crystals determined from CPI images are also listed for given time period.

| Period | Time (UTC) | $D_{max}$ (µm), $T$ (°C), altitude (km) | Single frozen droplet | Frozen droplet aggregates (FDAs) | Plate (PLT) | Column (COL) | Unclassified (UC) |
|---|---|---|---|---|---|---|---|
| All | 221100 – 222800 | 80.7±45.4, -58.1±1.4, 12.121±0.138 | 73.036 (40.014), 34.4±6.8 | 20.850 (46.308), 80.7±45.4 | 0.013 (0.059), 98.1±30.9 | 0.013 (0.022), 69.6±12.7 | 6.073 (13.539), 75.7±37.2 |
| 1 | 221100 – 221500 | 68.4±37.1, -60.0±1.2, 12.226±0.151 | 84.065 (65.691), 32.5±5.6 | 12.050 (27.786), 68.4±37.1 | – | – | 3.885 (6.523), 53.1±23.5 |
| 2 | 221900 – 222340 | 72.4±42.9, -57.5±0.3, 12.033±0.003 | 70.635 (39.354), 34.6±6.5 | 24.002 (49.615), 79.4±42.9 | – | – | 5.340 (10.922), 73.6±30.4 |
| 3 | 222350 – 222800 | 84.4±48.8, -56.6±0.5, 12.032±0.004 | 71.236 (35.423), 34.9±7.4 | 21.216 (47.467), 84.4±48.8 | 0.030 (0.115), 98.1±30.9 | 0.030 (0.043), 69.6±12.7 | 7.478 (16.922), 81.1±41.2 |



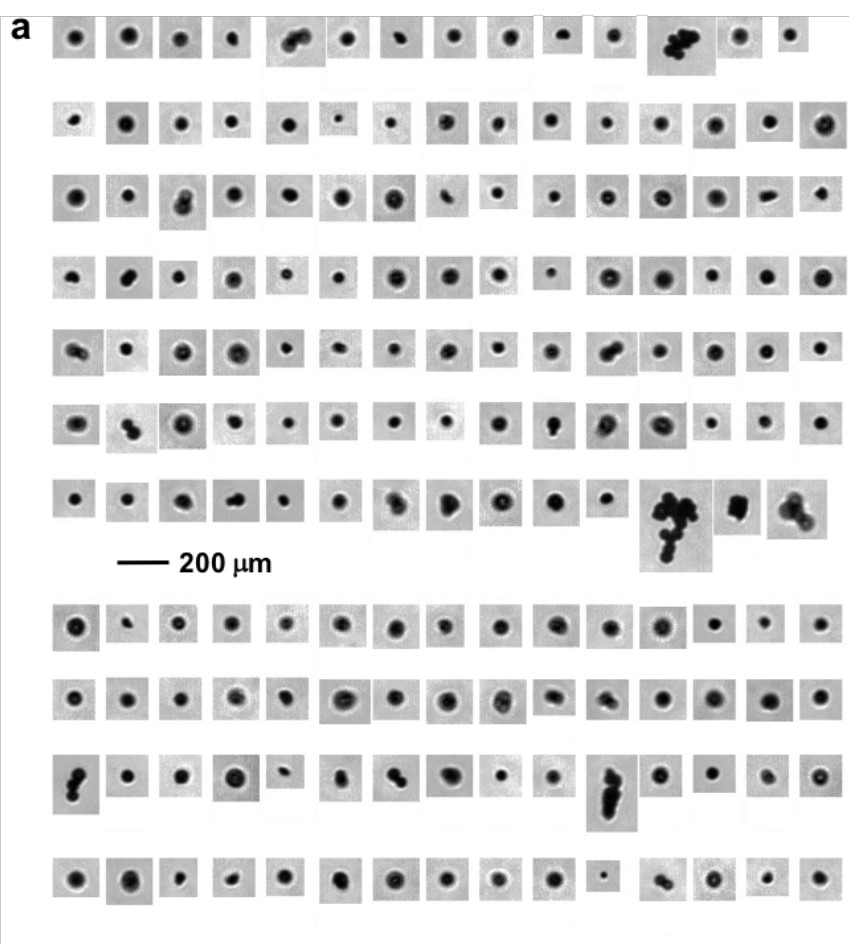



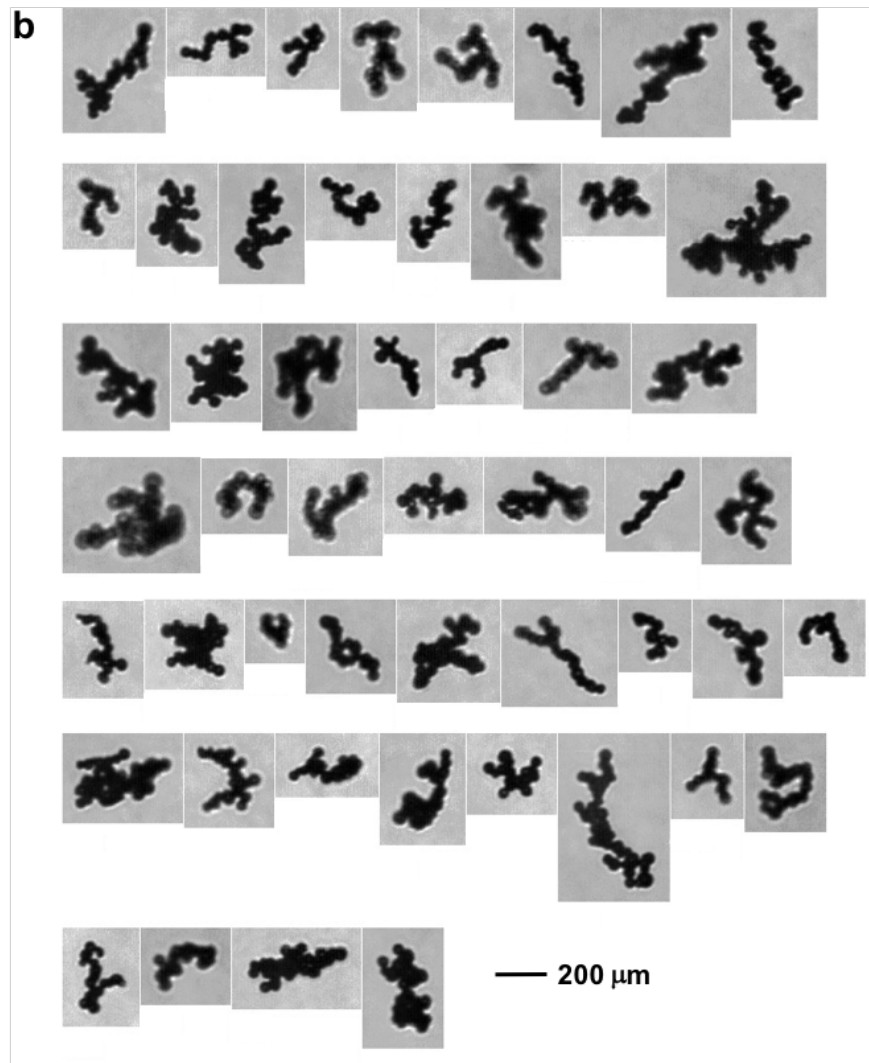

**Figure 1: (a) Example CPI images of ice crystals observed at $T$= –58.16 °C (altitude of 12.11 km) between 221213 and 221219 UTC, (b) example CPI images of ice crystals observed at $T$= –57.72 °C (altitude of 12.03 km) between 222102 and 222214 UTC. The 200 μm scale bar is embedded in each figure.**





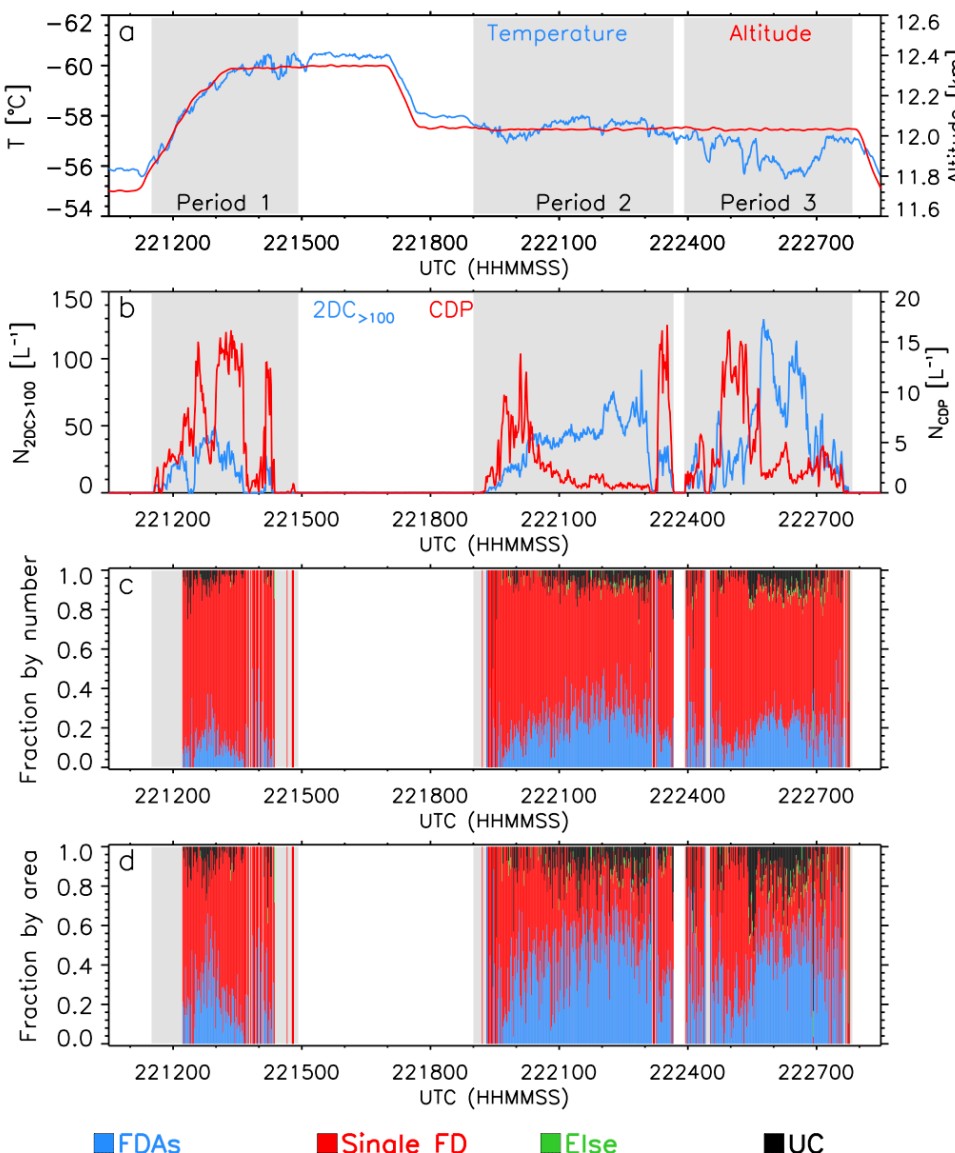

**Figure 2: (a) Temperature and altitude of NSF/NCAR G-V aircraft as a function of time on 6 June 2012 flight and (b) 1 s average concentration of measured by CDP and 2DC$_{>100}$. Determined habit fraction (c) by number and (d) by projected area as a function of time. Habit categories for panels (c) and (d) are shown at the bottom. Habits sorted into 15 categories: frozen droplet aggregates (FDAs), small quasi-sphere (SQS), medium quasi-sphere (MQS), large-quasi sphere (LQS), plate (PLT), aggregates of plates (APs), bullet rosette (BR), aggregates of bullet rosettes (ABRs), column**

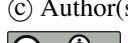

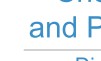

**(COL), aggregates of column (ACs), needle (ND), aggregates of needles (ANDs), dendrite (DEN), capped column (CC), and unclassified (UC) as shown in Fig. 2. For simplicity single frozen drop (FD) denoted in this figure includes SQS, MQS, and LQS, while other habits except FDAs and UC are indicated as Else in this figure. Time periods 1, 2, and 3 are shaded with gray color in each**

5   **panel.**





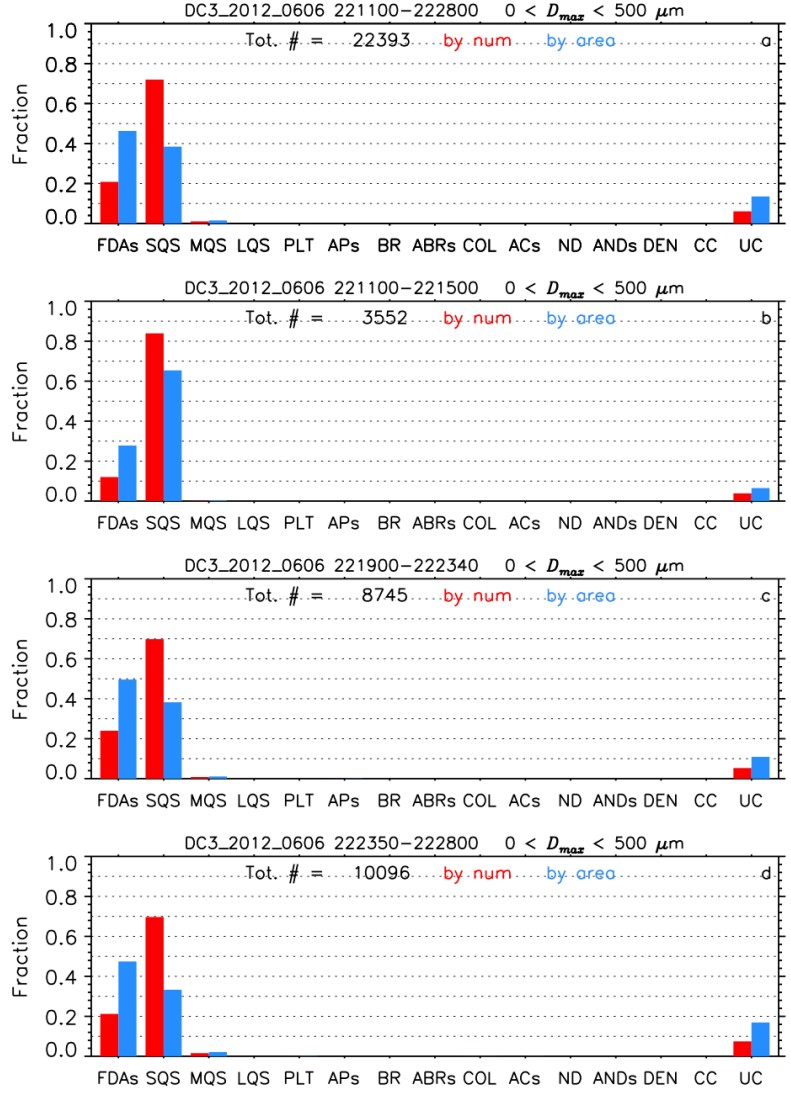

**Figure 3: Contributions of ice crystal habits by number (red) and by projected area (blue) during**

5   **(a) all periods, (b) period 1, (c) period 2, and (d) period 3. Total number of samples is indicated in**

**each panel. Acronyms for crystal habits are indicated in Fig. 2 caption.**



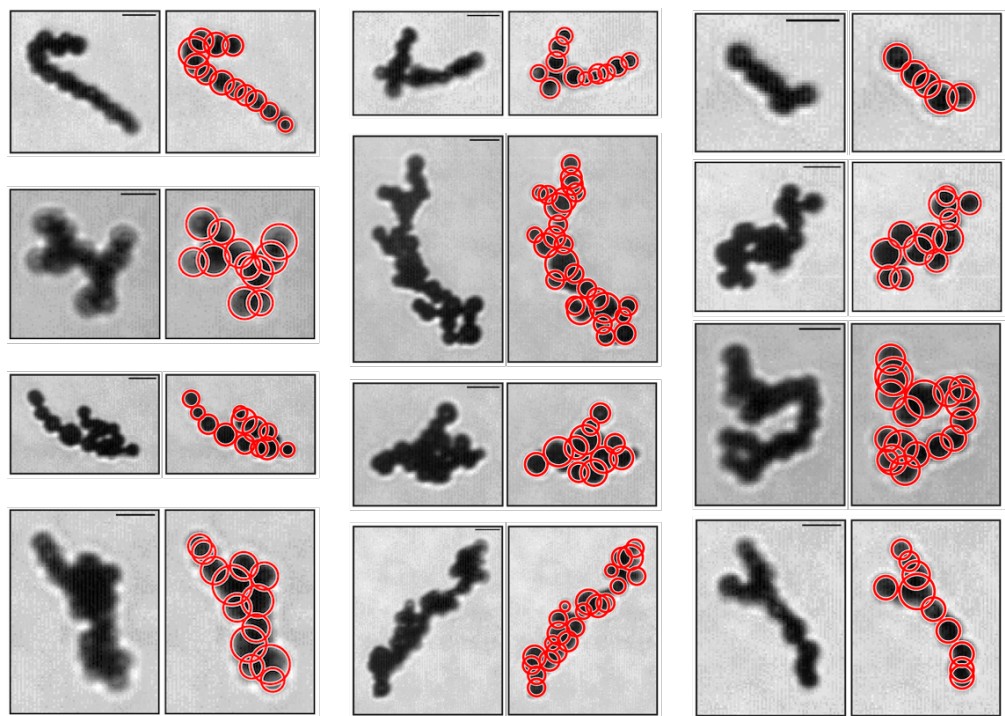

**Figure 4: CPI images of frozen droplet aggregates (FDAs, left image of each column) and those with determined element frozen droplet (red circle, right image of each column). The 46 µm scale bar is shown in each image.**





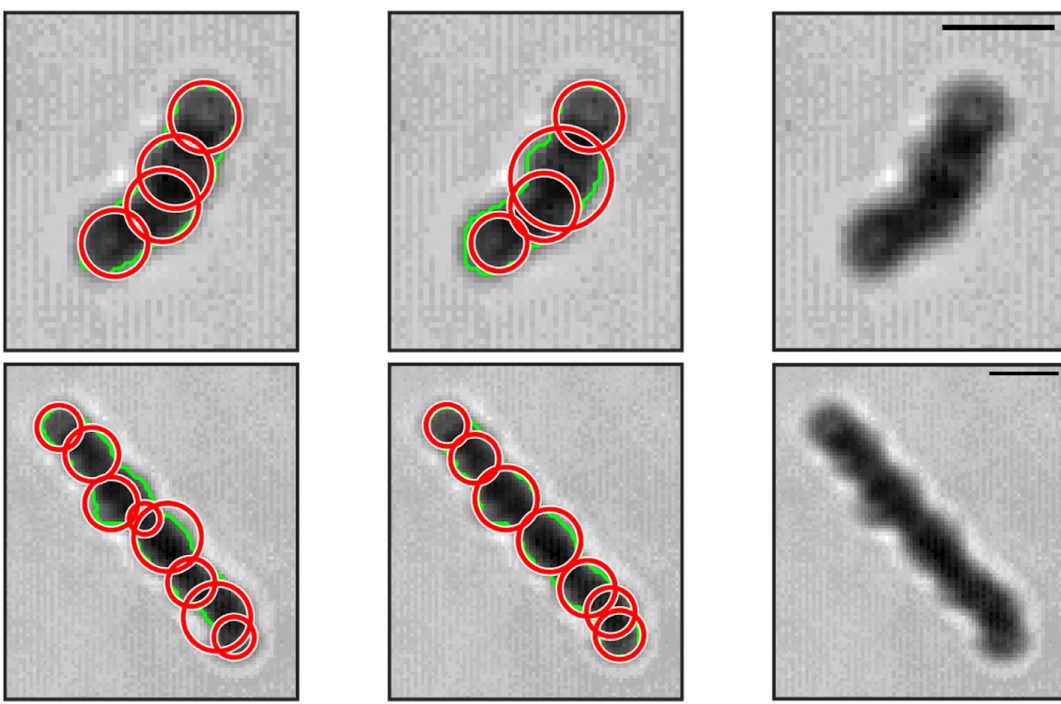

5    **Figure 5: Element frozen droplets (red circles) determined using phase-coding (left column) and two-stage CHT (middle column) techniques. Examples of two different FDAs are shown in top and bottom row, respectively. Original CPI images of FDAs are shown in right column with 46 μm scale bar. Detected boundary (green lines) of FDAs are shown on panels in left and middle columns.**



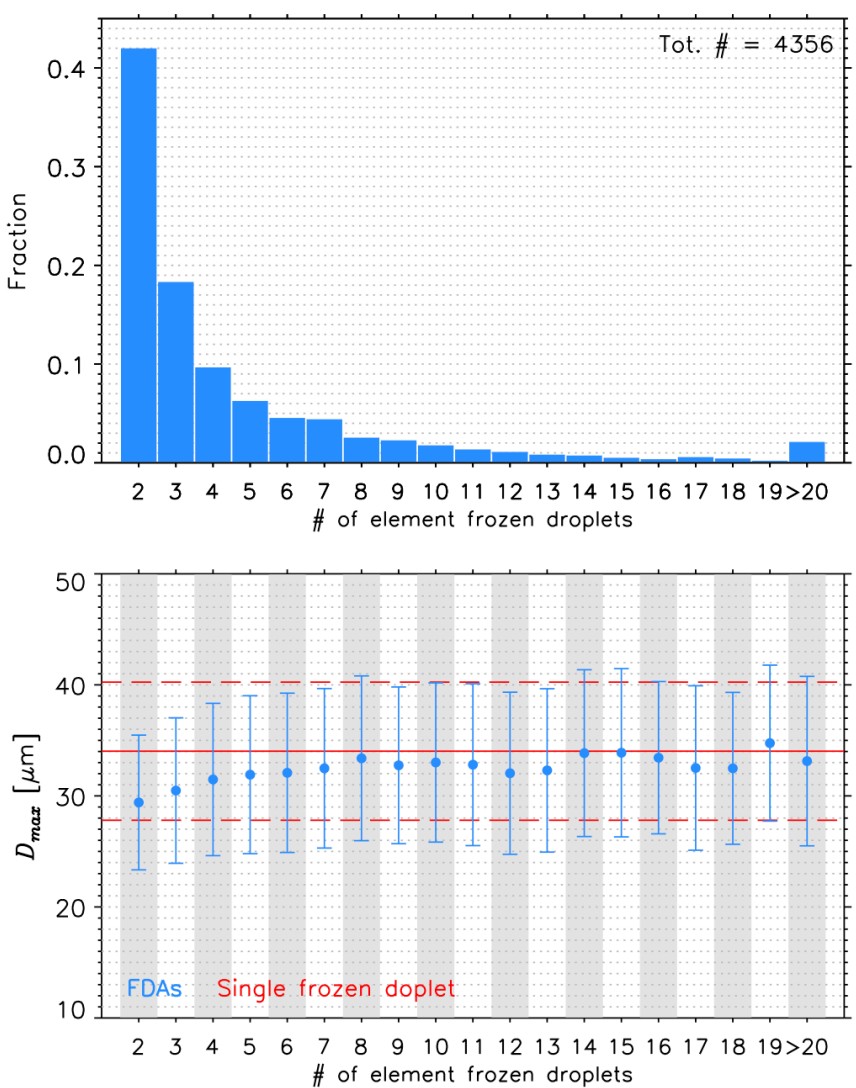

**Figure 6: (top) Normalized frequency of occurrence of number of component frozen droplet within**

5    **4,356 FDAs analyzed. (bottom) The average and standard deviation of diameter of frozen droplets**

**as a function of number of component frozen droplet within FDAs (blue). The average (red solid**

**line) and standard deviation (red dash line) of $D_{max}$ of single frozen droplets are also shown.**





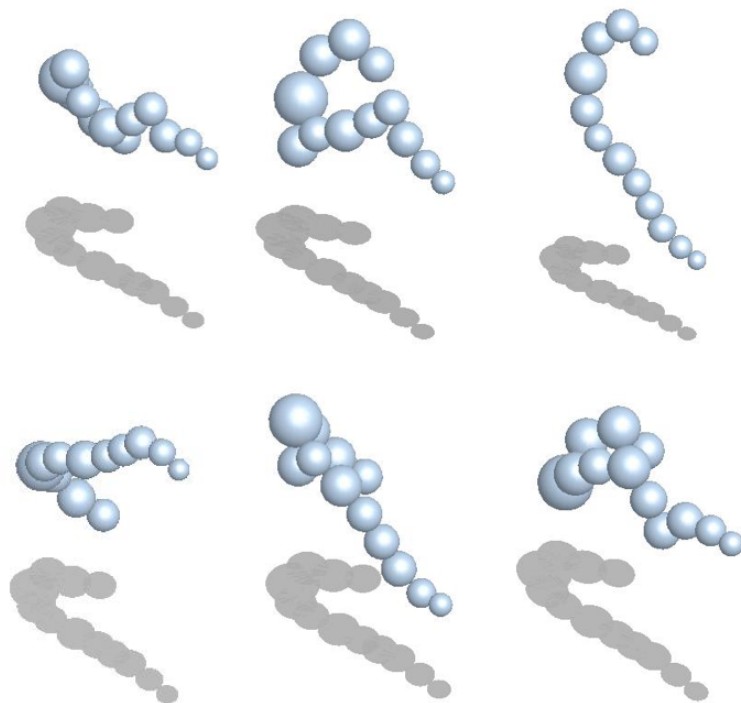

5    **Figure 7: Six different examples of three-dimensional representations of FDA. Each image has the same projected area (gray) that is the CPI image shown in Fig. 4 (top-left image).**





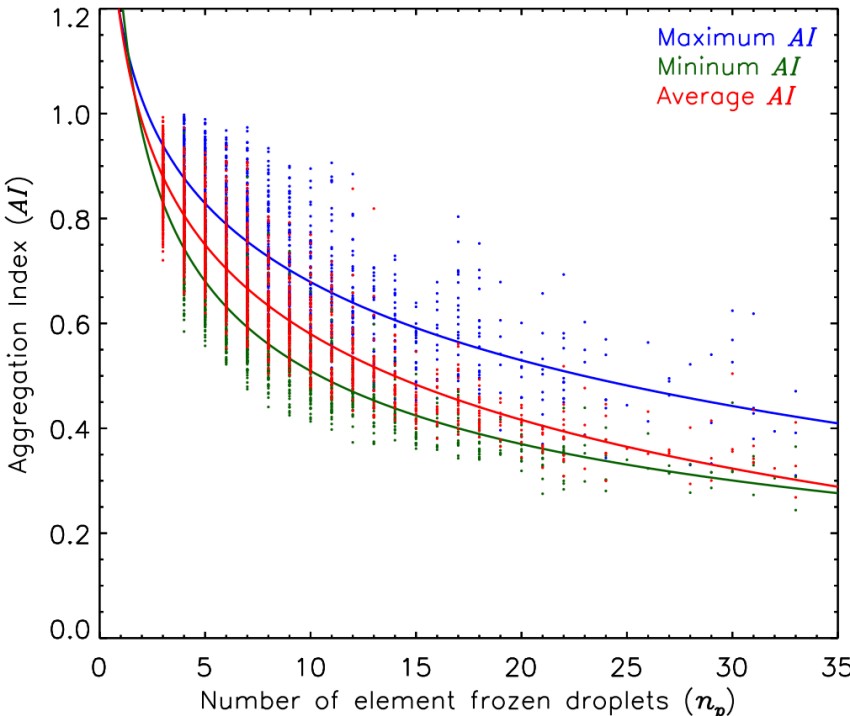

5    **Figure 8: The calculated maximum (blue), minimum (green), and average (red) aggregation index**
     **(*AI*) of FDAs (circles) as a function of the number of element frozen droplets. Best-fit lines are**
     **shown with solid lines.**



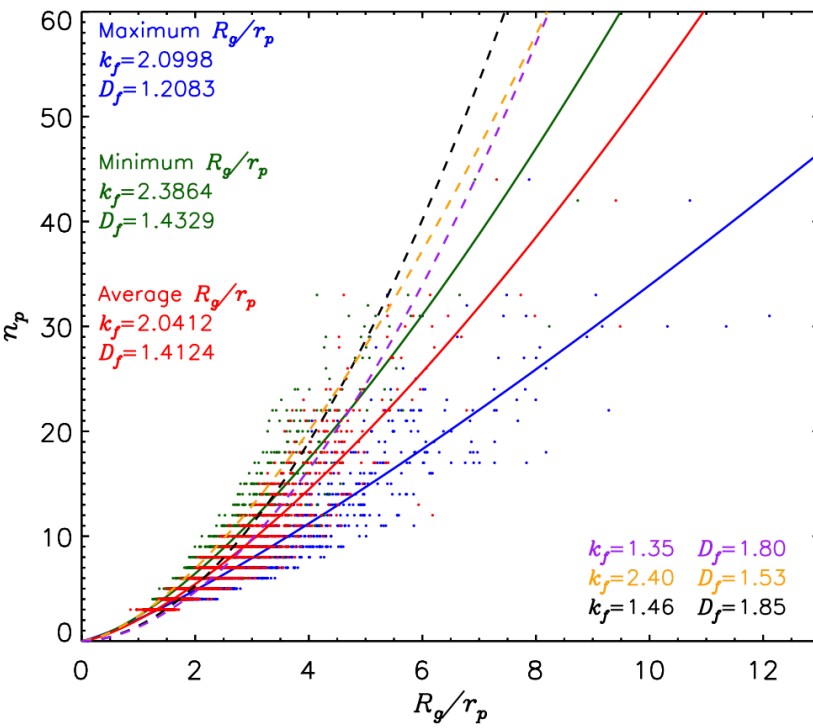

**Figure 9: Relationships between the ratio of radius of gyration ($R_g$) to average diameter of elements frozen droplets ($r_p$) and the number of element frozen droplets ($n_p$). Maximum (blue), minimum (green), and average (red) $R_g/r_p$ are shown with circles and corresponding best-fit lines are also plotted with solid lines. Calculated fractal dimension ($D_f$) and fractal prefactor ($k_f$) of FDAs with maximum, minimum, and average $R_g/r_p$ are embedded. The $D_f$ and $k_f$ of ambient (black) and denuded (yellow) black carbon aggregates determined in China et al., (2013) are indicated together with those commonly determined (purple) in several studies.**



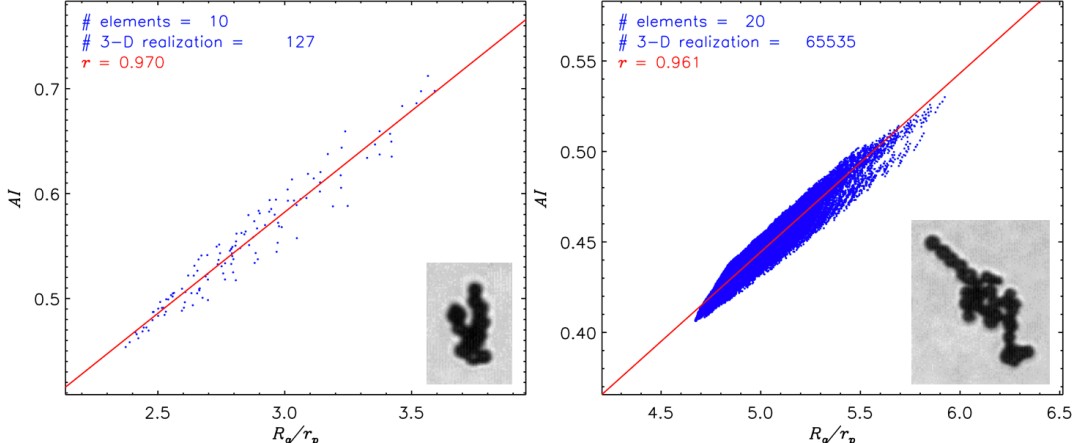

10 **Figure 10: Relationship between aggregation index($AI$) and the ratio of radius of gyration to average radius of elements ($R_g/r_p$) of FDAs. The number of element frozen droplets (# elements) and number of all possible 3-D realizations (# 3-D realization) are indicated. The best fit and correlation coefficient ($r$) for the relationship between $AI$ and $R_g/r_p$ are shown with red color. Corresponding CPI images of FDAs are also embedded in each panel.**

