# Peer review of "Microphysical Characteristics of Frozen Droplet Aggregates from Deep Convective Clouds"

_Atmospheric Chemistry and Physics, 2018_

## Referee Comment (RC1) · Anonymous Referee #1 · 14 Mar 2018

This is an innovative paper that obtains quantitative measures of the geometric properties of frozen droplet aggregates, demonstrated to dominate some cloud cases. The work will be useful in future studies of radiative properties and fall speeds of such particles. The paper is well presented and I only have some relatively minor comments, with the exception of the suggestion to publish it in AMT instead of ACP.

The circle Hough transform is an interesting new approach in this context, and it is worth publicising. However, the authors noted that the two sub-methods used (two-stage and phase-coding) performed differently for different aggregates, yet do not give any statistics, or suggest why they perform differently. Since the methods may lead to different bias, this should be elaborated on.

The authors say repeatedly that the relative frequency of occurrence of single frozen

droplets and FDAs depended on temperature and position within the anvil cloud. The same may be the case for the retrieved properties, like the AI or fractal dimension. Yet no quantitative data, even a simple as scatter plots or correlations are given. Why is this valuable information withheld? Without it, the paper is essentially a method paper, and appears to be more suitable for Atmospheric Measurement Techniques, not ACP.

It is not correct to assume idealised spheres as basic elements of the aggregates. This is because frozen droplets are not smooth spheres, and the scattering properties of chain aggregates tend towards the scattering properties of individual elements as the the chain straightens (due to diminishing interactions - this effect is seen e.g. in Um et al, 2009). Thus the scattering may become dominated by the low asymmetry parameter of rough spheres for some aggregate types. So for the future study that the authors propose in the Summary and Conclusions microscopic detail of the particle surface must be taken into account, determining the selection of the single-scattering modelling method, e.g. superposition T-matrix would not be appropriate.

Minor correction: p.1, line 28: this sentence needs rewriting.

---

## Referee Comment (RC2) · Anonymous Referee #2 · 13 Apr 2018

**1   Main Comments**

The paper analyses 22k+ images of ice crystals within the upper anvil region of two storms over Colorado. They apply a circle Hough transform technique to identify the positions of element-frozen droplets, which appears to be a new and useful technique in this context.

In the abstract they talk about the relative frequency of occurrence of different habits depending on temperature and position within the anvil cloud. While this may be expected I did not see any results that supported this statement.

A statement that the CPI error in sizes is +/- 4.6 microns is given on page 9, line 15. I

think this may underestimate the errors when particles are positioned away from best focus. What value of the focus parameter from the CPI processing software was used? Particles around 30 microns diameter can be oversized by a factor of 1.6 due to these effects.

Overall the techniques used and results presented are of a high quality; however, if the paper is to be published in ACP I feel more should be made of the relative abundance of single frozen and FDAs in these anvil clouds – this is the main scientific finding of the paper. Why is it important to know these, and how can the measurements / findings be used by modellers, etc. Could the aggregation indices in figure 5 help modellers understand the nature of the aggregation process for instance? i.e. whether electric fields are important, or whether it is more random. I think this may be the case. It may be worth presenting a discussion about the electric field-aggregation hypothesis, and then presenting these statistics as a way of testing it?

The methodology is innovative and very useful; however, when presented alone it would warrant publication in a techniques paper like AMT.

**2 Specific comments**

- BC not defined in the abstract. Assume you mean Black Carbon.

- Woodely should be Woodley page 3, line 18.

- Figure 2: a complicated habit recognition scheme is described in the figure caption, but the acronyms do not appear in the figure

---

## Author Comment (AC1) · 3 Sep 2018

**Reviewer #1**

This is an innovative paper that obtains quantitative measures of the geometric properties of frozen droplet aggregates, demonstrated to dominate some cloud cases. The work will be useful in future studies of radiative properties and fall speeds of such particles. The paper is well presented and I only have some relatively minor comments, with the exception of the suggestion to publish it in AMT instead of ACP.

We thank the reviewers for their careful reviews and constructive suggestions.

The circle Hough transform is an interesting new approach in this context, and it is worth publicising. However, the authors noted that the two sub-methods used (twostage and phase-coding) performed differently for different aggregates, yet do not give any statistics, or suggest why they perform differently. Since the methods may lead to different bias, this should be elaborated on.

*"Two different techniques are used because the performance of each technique varies depending on the CPI image being classified. The technique used for the subsequent analysis is chosen as that for which the projected area of the FDAs determined for the element frozen droplets identified by CHT technique (i.e., area determined by red lines in Fig. 5) best matches that for the original CPI image (i.e., area enclosed by green line in Fig. 5). For example, the phase-coding technique shows closer agreement to the imaged area for the FDAs shown in the top row of Fig. 5, while the two-stage CHT does for the FDAs shown in the bottom row."*

Since both techniques showed similar performance we used both methods. Following original sentences (italic) shown above we added sentences as the reviewer suggested.

"But, the performance of both techniques is quite similar.".………………"Although the phase-coding method provided marginally better results for ~54% cases, the differences in projected area determined by two techniques were within 9.7% for all FDAs. Thus, since the performance of both methods in replicating the determined area of the CPI images is similar, there should be minimal bias in the determined results."

The authors say repeatedly that the relative frequency of occurrence of single frozen droplets and FDAs depended on temperature and position within the anvil cloud. The same may be the case for the retrieved properties, like the AI or fractal dimension. Yet no quantitative data, even a simple as scatter plots or correlations are given. Why is this valuable information withheld? Without it, the paper is essentially a method paper, and appears to be more suitable for Atmospheric Measurement Techniques, not ACP.

Thank you for pointing this out. We did further microphysical analysis and a new figure (Fig. 9 in revised manuscript) has been added. Figure 9 shows the dependence of AI on temperature. It shows that the AI of FDAs decreases with increasing temperature. This indicates that larger and more compact-shape FDAs exist in lower regions (i.e., higher temperature) of upper anvils.

We feel that this manuscript is more suitable for ACP than AMT for several reasons. First, only Section 3.2 is about the newly developed technique and other sections (more than 90% of this manuscript) are about analysis of microphysical properties of frozen droplet and their aggregates in upper anvil clouds. Second, more microphysical analysis has been added in the revised manuscript as the reviewer suggested. Third, all similar previous studies (Baran et al., 2012; Gayet et al., 2012; Stith et al., 2014; Stith et al., 2016) on this subject (i.e., observations of FDAs) were published in ACP.

It is not correct to assume idealised spheres as basic elements of the aggregates. This is because frozen droplets are not smooth spheres, and the scattering properties of chain aggregates tend towards the scattering properties of individual elements as the the chain straightens (due to diminishing interactions - this effect is seen e.g. in Um et al, 2009). Thus the scattering may become dominated by the low asymmetry parameter of rough spheres for some aggregate types. So for the future study that the authors propose in the Summary and Conclusions microscopic detail of the particle surface must be taken into account, determining the selection of the single-scattering modelling method, e.g. superposition T-matrix would not be appropriate.

We agree with the reviewer's point of view. Frozen droplets do not have spherical shapes, but rather quasi-spherical shapes that are important for the radiative property calculations. However, since this study identified the number, size, and relative position of element frozen droplets within FDAs the assumption of spherical shape is still valid. For example, an identifying number, size, and relative position of element frozen droplets within FDAs does not depend on an assumption of spherical or spheroidal shape element.

Minor correction: p.1, line 28: this sentence needs rewriting.

The sentence has been revised.

---

## Author Comment (AC2) · 3 Sep 2018

**Reviewer #2**
**1 Main Comments**

The paper analyses 22k+ images of ice crystals within the upper anvil region of two storms over Colorado. They apply a circle Hough transform technique to identify the positions of element-frozen droplets, which appears to be a new and useful technique in this context.

We thank the reviewers for their careful reviews and constructive suggestions.

In the abstract they talk about the relative frequency of occurrence of different habits depending on temperature and position within the anvil cloud. While this may be expected I did not see any results that supported this statement.

Different habits indicate single frozen droplet, frozen droplet aggregates, and other habits in this study. Comparing against the frequency of occurrence of single frozen droplet and frozen droplet aggregates, the frequency of occurrence of all other habits are very low (~7% by number, Table 1 and Figs. 2 and 3) as shown in Section 4.1.

A statement that the CPI error in sizes is +/- 4.6 microns is given on page 9, line 15. I think this may underestimate the errors when particles are positioned away from best focus. What value of the focus parameter from the CPI processing software was used? Particles around 30 microns diameter can be oversized by a factor of 1.6 due to these effects.

We used the focus > 20 following McFarquhar et al. (2013). The CPI used during the DC3 is 3V-CPI that has improved optical characteristics compared with those in Conolly et al. (2007) and McFarquhar et al. (2013). Based on the reviewer's suggestion, we added following sentence and corresponding reference.

"The CPI errors in sizing particle vary with focus (Connolly et al., 2007) and can be larger than those considered in this study."

Conolly, P. J., Flynn, M. J., Ulanowski, Z., Choularton, T. W., Gallagher, M. W., and Bower, K. N.: Calibration of cloud particle imager probes using calibration beads and ice crystal analogs: The depth of field, J. Atmos. Ocean. Tech., 24, 1860-1879, 2007.

McFarquhar, G. M., Um, J., and Jackson, R.: Small cloud particle shapes in mixed-phase clouds, J. Appl. Meteorol. Clim., 52, 1277–93, 2013.

Overall the techniques used and results presented are of a high quality; however, if the paper is to be published in ACP I feel more should be made of the relative abundance of single frozen and FDAs in these anvil clouds – this is the main scientific finding of the paper. Why is it important to know these, and how can the measurements / findings be used by modellers, etc. Could the aggregation indices in figure 5 help modellers understand the nature of the aggregation process for instance? i.e. whether electric fields are important, or whether it is more random. I think this may be the case. It may be worth presenting a discussion about the electric field-aggregation hypothesis, and then presenting these statistics as a way of testing it? The methodology is innovative and very useful; however, when presented alone it would warrant publication in a techniques paper like AMT.

The results of this study (i.e., derived AI) have important implication to improve the calculations of the microphysical (e.g., fall velocity) and radiative (e.g., asymmetry parameter) properties of ice crystals in upper anvil clouds, especially continental convective clouds. It may take time to implement the results of this study in numerical models and remote sensing algorithm, but it worthwhile to do as stated in the Summary and Conclusion section. We also added following sentences at the end of manuscript:

"To implement the results of this study for numerical models and satellite-retrieval algorithms, the role of electric fields within clouds should be identified and quantified systemically. Recent laboratory experiment (Pedernera and Ávila, 2018) showed that the collision and adhesion process was highly related to electrical forces that stimulated aggregation process of frozen droplet aggregates."

Pedernera, D. A. and Ávila, E. E.: Frozen-droplets aggregation at temperature below –40 °C. J. Geophys. Res., 123, 1244-1252, 2018.

We feel that this manuscript is more suitable for ACP and still want this manuscript to be published in ACP because of several reasons. First, only Section 3.2 is about the newly developed technique and other sections (more than 90% of this manuscript) are about analysis of microphysical properties of frozen droplet and their aggregates in upper anvil clouds. Second, more microphysical analysis (Figure 9 and the corresponding explanation) has been added in the revised manuscript as the reviewer suggested. Third, all similar previous studies (Baran et al., 2012; Gayet et al., 2012; Stith et al., 2014; Stith et al., 2016) on this subject (i.e., observations of FDAs) were published in ACP.

**2 Specific comments**

 • BC not defined in the abstract. Assume you mean Black Carbon.

      "Black carbon" has been added in the abstract.

 • Woodely should be Woodley page 3, line 18.

      It has been revised.

 • Figure 2: a complicated habit recognition scheme is described in the figure caption, but the acronyms do not appear in the figure

      Unnecessary acronyms in Fig. 2 have been removed as the reviewer indicated.

---

## Author Comment (AC3) · 3 Sep 2018

[Figure]

Figure 9: The average aggregation index (AI, red circles in Fig. 8) as a function of temperature (blue circles). The mean and standard deviation of *AI* for six temperature ranges are indicated with red circles and vertical bars.

---

## Referee Report (RR1)

**Microphysics characteristics of frozen droplet aggregates from deep convective clouds**

This paper presents an analysis of ice particle images from an anvil cloud. Fractal dimensions and other parameters are calculated from the raw images by using the Hough transform technique. It is notable that the images are frozen-droplet aggregates as this has implications to improve fall-velocity estimations and radiative transfer calculations in anvil clouds. As this is the second iteration I have focussed on the results and discussion / conclusions, rather than the material prior.

I only have a few comments, which may improve clarity of the manuscript – so that others may repeat similar analyses

Your analysis assumes that the maximum number of contacting points of an element frozen droplet with other frozen droplets is 2. How valid is this assumption, since there is little justification in the manuscript. Looking at figure 1a, there maybe some examples where this is not true, but it is hard to tell. A comment on this point would help.

Stacking permutations: is described as $2^{(n-2)}$. Is this the case for droplets of different sizes, as in your case? Looking at figure 6 and 7 it appears that the droplet elements have *different* sizes, so I am not sure whether this stacking assumption still applies? I do not think it affects your results, but it confused me for a while as to what was actually done.

I have no other problem with the methods used. I am convinced by the method, and by number of calculations done. I also feel like the choice of parameters used to characterise the aggregates is good.

Do you have a feel for how commonly applicable the results are – because this is data from just one flight. It may be worth a comment if you do, or maybe a comment like "this analysis needs to repeated for other datasets to assess how widely applicable they are"